# Evening Chronotype and Suicide: Exploring Neuroinflammation and Psychopathological Dimensions as Possible Bridging Factors—A Narrative Review

**DOI:** 10.3390/brainsci14010030

**Published:** 2023-12-28

**Authors:** Luca Magnani, Andrea Aguglia, Jacques Alexander, Alessandra Maiorano, Hélène Richard-Lepouriel, Sidonia Paula Iancau, Andrea Amerio, Alberto Parise, Gianluca Serafini, Mario Amore, Khoa D. Nguyen, Alessandra Costanza

**Affiliations:** 1Department of Psychiatry, San Maurizio Hospital, 39100 Bolzano, Italy; magnani1991@gmail.com; 2Department of Neuroscience, Rehabilitation, Ophthalmology, Genetics, Maternal and Child Health, Section of Psychiatry, University of Genoa, 16132 Genoa, Italy; andrea.aguglia@unige.it (A.A.); andrea.amerio@unige.it (A.A.); gianluca.serafini@unige.it (G.S.); mario.amore@unige.it (M.A.); 3IRCCS Polyclinic Hospital San Martino, 16132 Genoa, Italy; 4Department of Psychiatry, Geneva University Hospital (HUG), 1205 Geneva, Switzerland; jacques.alexander@hcuge.ch (J.A.); alessandra.maiorano@hcuge.ch (A.M.); 5Department of Psychiatry, Mood Disorder Unit, Psychiatric Specialties Service, Geneva University Hospital (HUG), 1205 Geneva, Switzerland; helene.richard-lepouriel@hcuge.ch; 6Residence School in Psychiatry, Faculty of Medicine and Psychology, Sant’Andrea Hospital, Sapienza University, 00185 Rome, Italy; sidoniapiancau@gmail.com; 7Geriatric-Rehabilitation Department, University Hospital of Parma, 43126 Parma, Italy; aparise@ao.pr.it; 8Chinese University of Hong Kong, Hong Kong SAR, China; khoa.d.nguyen@gmail.com; 9Tranquis Therapeutics, Palo Alto, CA 94303, USA; 10Department of Psychiatry, Faculty of Biomedical Sciences, University of Italian Switzerland (USI), 6900 Lugano, Switzerland; 11Department of Psychiatry, Faculty of Medicine, Geneva University (UNIGE), 1205 Geneva, Switzerland

**Keywords:** chronotype, eveningness, E-chronotype, social jet leg, mood dysregulation, seasonality, depression, suicide, suicidal ideation, suicidal behavior, inflammation, neuroinflammation

## Abstract

A chronotype is generally defined as the variability of the phase angle of entrainment, while the latter reflects the relationship between the timing of a certain rhythm (e.g., the sleep–wake cycle) and the timing of an external temporal cue. Individuals can be placed on a spectrum from “morning types” (M types) to “evening types” (E types). E-chronotype has been proposed as a transdiagnostic risk factor for psychiatric conditions, and it has been associated with psychopathological dimensions. Eveningness seems to be correlated with both suicidal ideation (SI) and suicidal behavior (SB) through several possible mediating factors. Immunological alterations have also been linked to later chronotypes and SI/SB. This narrative review aims to summarize the evidence supporting the possible association between chronotypes and suicide and the eventual mediating role of neuroinflammation and several psychopathological dimensions. A search of the literature (2003–2023) was conducted using various databases: PUBMED, EMBASE, Scopus, UpToDate, PsycINFO, and Cochrane Library. English-language articles were collected and screened for eligibility. Despite the apparent absence of a direct correlation between E-chronotype and suicidality, E-chronotype promotes a chain of effects that could be involved in an increased risk of SB, in which with neuroinflammation possibly plays an intriguing role and some psychopathological dimensions may stand out.

## 1. Introduction

From an ecological perspective, living organisms exist both in spatial and temporal niches [1]. Within each different spatiotemporal environment, certain regularities can be identified and used by different organisms to develop better forms of adaptation [1]. Therefore, selective pressures have always acted to facilitate the expression of those biological traits that are most associated with the aforementioned adaptive tendency, which also refer to regularities in the temporal architecture of the environment. In this sense, dampened or self-sustained internal biological rhythms represent a refined evolutionary product [1]. Humans, as complex living organisms, express different biological rhythms (ultradian, circadian, infradian) from the level of the individual cell to that of general behavior according to different biological clocks [1]. Specifically, for circadian rhythms, the suprachiasmatic nucleus (SCN) represents a fundamental center of multilevel orchestration within the central nervous system (CNS) in mammals [2].

Within this general and evolutionary framework, the chronotype can be defined as the individual variability in the phase angle of entrainment [3,4]—that is, the relationship between the timing of a certain internal biological clock (with its linked physiological and behavioral manifestations) and the timing of an external temporal cue (i.e., the *zeitgeber*), which is particularly capable of affecting the course of the associated biorhythm. However, chronotypes usually refer to sleep–wake behaviors in humans [5], and light exposure during day–night cycles thus represents the main *zeitgeber* [6,7]. In any case, the expression of behavioral chronotypes is not an exclusively human prerogative [4], which again reflects its importance in evolutionary terms.

When referring to sleep–wake behaviors, the chronotype can be indirectly assessed by checking the so-called mid-sleep point on free days (MSF)—i.e., with no alarm clock. The MSF can be defined, for instance, through questionnaires, such as the Munich ChronoType Questionnaire (MCTQ) [8], and it should subsequently be corrected for oversleeping (MSF-sc). Apparently, chronotype expression estimated in this way takes the form of a near-Gaussian distribution [9]. Consistently and according to circadian typology [10,11], individuals are usually placed on a spectrum ranging from extreme “morning types” (M types) to extreme “evening types” (E types), with most subjects falling into the wide category of “neither type” (N types). The expression of chronotypes as defined above seems to be influenced by a group of constitutive and environmental factors (Table 1).

The variations of many relevant variables have been investigated in association with more defined chronotypes (M or E types). Important findings emerged regarding, for example, body temperature [24], cortisol [25] and melatonin [26] secretion, metabolic parameters [27], cognitive abilities [28], and personality traits [11,29]. Despite some well-known changes occurring during adolescence and young adulthood [15,30], the chronotype is generally considered a biologically rooted and relatively stable state (i.e., trait-like character) [31], which nevertheless tries to encapsulate the dynamic qualities of the phase of entrainment [7].

In recent years, many efforts have been made to investigate human chronotypes in association with physical and mental health [32]. In the paragraphs below, we illustrate how the late chronotype, favoring the so-called circadian disruption, may play a role as a transdiagnostic risk factor in many psychopathological conditions, some of which are known to be related to SI/SB. In addition, we try to further unfold the link between E-chronotype and SI/SB by focusing on neuroinflammation because eveningness appears to promote higher levels of local and systemic inflammation [33,34,35,36,37]. Thus, this narrative review aims to summarize evidence supporting the possible association between chronotypes and suicide by considering a complex causal architecture within which the mediating role of neuroinflammation and some psychopathological dimensions may stand out.

## 2. Materials and Methods

We performed a narrative review. An extensive and nonsystematic search of the most relevant and recent literature (2003–September 2023) was conducted using the following databases: PUBMED, EMBASE, Scopus, UpToDate, PsycINFO, and Cochrane Library. English-language articles were identified, collected, and screened (title–abstract level and full-text level) for eligibility by two different reviewers while adopting expansive inclusion criteria with the aim of providing a general yet quite complete perspective on the chosen topic. The search strategies utilized index terms, where appropriate, and free text terms to capture the relevant concepts and the associations between chronotypes, social jetlag, psychopathology, immune dysregulation, and suicide. The results are finally presented in a way that tries to explain the possible link between SI/SB and later chronotypes in a progressive way.

## 3. Results

### 3.1. E-Chronotype and Social Jetlag

Jetlag is commonly defined as a transient misalignment between a person’s circadian rhythm and the local timing [38]. We specifically refer to social jetlag (SJ) as a condition of misalignment between the preferred and socially required timing of a certain daily activity; for instance, the difference between sleep timing on workdays and the weekend [7]. When chronically experienced, SJ may lead to a condition called “circadian disruption” [7,39,40]. Unlike the notion of SJ, the definition of circadian disruption entails a range of biological effects across several scales of observation (from molecules to cells, tissues, and the organism as a whole), suggesting its possible association with a complex of psychophysical consequences. In line with previous studies [41], the Spanish study conducted by Martinez-Lozano et al. [42] investigated whether E-chronotype was linked with a pattern of psychophysical effects in school-age children. In a sample of 432 children aged 8–12, the chronotype was measured objectively (seven days of wrist temperature, physical activity, and body position) and subjectively (MCTQ). The measure was then integrated into information about sleep rhythms and light exposition, food intake, anthropometric data, metabolic parameters, and academic scores. E-type children, compared to M-type ones, more frequently presented SJ (7% and 3%, respectively; *p* = 0.001), sleep alterations, including, in particular, a lower sleep circadian function index (*p* = 0.007) with decreased relative amplitude and lower interday stability, obesity risk, and as a higher metabolic risk. Importantly, in a recent systematic review including seven studies, Henderson et al. [43] investigated the association between SJ and mental health outcomes in young people. Although the findings appeared equivocal, Henderson et al. found associations between SJ and clinical and seasonal depression, but only in female participants living in higher-latitude regions [40,43]. More recently, interesting results have come from other works. Chen et al. [26] showed that 46.9% of a sample of 4787 adolescents had SJ and more behavioral difficulties (OR: 1.20–1.34, *p* = 0.03; 0.02), even after controlling for several factors. A dose–response relationship emerged with the risk of conduct problems and hyperactivity [44]. An SJ of ≥1 h was significantly associated with elevated risk of irritable mood in a study by Tamura et al. [45] conducted on 4782 students (12–15 years). Therefore, chronic SJ (i.e., circadian disruption) appears to be a possible first piece of a complex association between chronotypes and further psycho-behavioral effects [46]. As seen, adolescents and young adults naturally tend to shift their circadian rhythm to that of an E-chronotype, while school/work timing is usually set to early waking time. Therefore, as suggested by the above evidence, the prevalence of SJ and circadian disruption seems to increase from childhood to adolescence/young adulthood, with all of their potential consequences.

### 3.2. E-Chronotype and the Expression of Psychopathological Dimensions

E-chronotype may be a predictor of the expression of many psychopathological dimensions both during the prodromal/subacute/chronic disease phases and the overt/acute episodes. For example, eveningness appears to be associated with different aspects of attention deficit hyperactivity disorder (ADHD) symptomatology [46]. More specifically, the link is particularly pronounced for the inattentive dimension, as a correlation with impulsivity–hyperactivity (externalizing behavior) was not found in a sample of 205 adults tested with the Self-Report Scale for ADHD (ASRS) and the Composite Scale of Morningness (CSM) [47]. However, a review of 13 studies by Schlarb et al. [48] illustrates how externalizing behavior could also be chronotype-related because the circadian rhythm seems to be connected to emotional regulation. In fact, eveningness emerged as associated with behavioral problems, such as aggression and impulsivity, in adolescents and children. Social anxiety and emotional dysregulation appeared to be negatively associated with morningness (*p* = 0.01) in a sample of 510 students with no history of psychiatric disorders [49]. Insomnia and sleep problems have been also linked to E-chronotypes [50,51,52,53,54].

Endogenous abnormalities of the circadian rhythm, SJ, and circadian disruption are also hypothesized to be central mechanisms in the pathophysiology of Bipolar Spectrum Disorders (BSDs), as circadian delay may provide a first vulnerability element for BSD onset. Moreover, chronotypes have also been directly associated with manic–depressive symptoms. E-chronotype and circadian delay in BSD patients were particularly associated with the depressive phase [55]. Later chronotypes in BSD patients involved in a 5-year follow-up study (*n* = 318) led to experiencing fewer hypomania/mania episodes, but these patients were found to be more likely to have mild to more severe depressive symptoms over the follow-up period [56]. At the same time, a 2015 case-control study by Pirkola et al. [57] compared a small group of patients affected by BSDs (*n* = 16, among them 12 affected by Bipolar Disorder I (BD I), 1 by Bipolar Disorder II (BD II), 2 by cyclotymia, and 1 by schizoaffective disorder) with their healthy relatives; mood symptoms were assessed with the Mood Disorder Questionnaire (MDQ) and chronotype was assessed with the Morningness–Eveningness Questionnaire (MEQ). The MDQ and MEQ scores had an inverse correlation (*p* = 0.016). The results also indicated increased lifetime manic behavior among the E-chronotypes. Despite seemingly conflicting evidence, the E-chronotype seems reasonably associable with an increased susceptibility to mood destabilization and emotional dysregulation, and this may properly represent the fundamental vulnerability element for the various BSD manifestations. Further exploring seasonal cofactors, Lewy’s phase shift hypothesis aims to explain how lowered mood in the winter may be associated with phase delay towards eveningness while clarifying that this aspect may be important not just in clinical cases of seasonal affective disorder but also in less severe mood swings [58]. Interestingly, a significant association (*p* < 0.01) was found in a random community sample of 335 non-psychiatric subjects between this specific circadian rhythm variation and mood disorder during winter [59], and evening types among a sample of 1539 adolescents had significantly higher seasonality scores than M-types (*p* < 0.001), even when measured prospectively over time [60]. Thus, seasonality appears to be another dimension at least partially influenced by chronotypes.

Still regarding mood, the available evidence concerns not only correlations between mood swings, depressive, and manic/hypomanic episodes in bipolar patients and E-chronotype, but also between the latter and unipolar depressive manifestations. A metanalysis conducted by Au et al. [61] involving 36 studies (*n* = 15,734) correlates eveningness to greater depressive symptom intensity both in clinical and subclinical populations, although with an overall small effect size. The association between E-chronotype and a greater likelihood of depression has been observed both in individuals with Major Depressive Disorder (MDD) and even in healthy controls [52,62,63,64]. Because depression is considered a leading cause of SI/SB, it may represent an advanced mediating factor between late chronotype and SI/SB, and it is also possibly linked to other neurobiological correlates, such as systemic or local inflammation (see further).

### 3.3. E-Chronotype as a Transdiagnostic Risk Factor for a Group of Psychiatric Conditions

Promoting the individual overexpression of some psychopathological dimensions, the E-chronotype may represent a transdiagnostic risk factor for the emergence of a group of psychiatric conditions, as apparently confirmed by some evidence. A further link between chronotype and depression has been examined in a Finnish study [64] involving 6071 participants; evening types reported depressed mood and loss of interest more frequently than morning types, thus distinguishing themselves as more prone to depression. In other works, the later chronotype has been associated with more hard-to-treat depressive conditions [65] and with a greater likelihood of non-remission of depression (OR 3.36; 95% CI 1.35–8.34; *p* < 0.01), even when controlling for insomnia severity [50]. According to some evidence, people with BD (both type I and II) are more likely to be an E-chronotype than healthy controls [66,67,68,69], even when they are euthymic [70]. Moreover, eveningness appeared to be a relatively stable characteristic irrespective of mood state when assessed longitudinally in BSD subjects [67,71]. In a case-control study involving 190 adults affected by BSDs and 128 age/gender/ethnicity-matched healthy controls living in the same geographical area, the evening types were more frequent in the BSDs group [67]. In another case-control study, bipolar patients were more frequently identified as evening types than schizophrenic/schizoaffective patients and controls [66]. Importantly, in the same work, younger type BD I patients and those with rapid mood swings were significantly more likely to have lower composite scale scores (i.e., to score in the “evening” range and to have a later circadian phase). Analogue results have been shown in a more recent case-control Korean study evaluating chronotype with CSM and involving 92 patients with BD I, 113 with schizophrenia (SZ), and 95 controls [68].

E-chronotype does not appear to be related only to affective or mood disorders. In fact, eveningness seems to also be linked to alterations in reward function, possibly leading to abnormal behaviors of sensation seeking and substance use. Through a monetary reward paradigm, a study conducted by Hasler et al. [72] analyzed different images obtained through functional magnetic resonance in two regions of interest (ROI: medial prefrontal cortex (mPFC), ventral striatum (VS)) associated with neural response to reward. E-types (*n* = 21), compared to M-types (*n* = 13), showed decreased mPFC responsiveness during reward anticipation and increased activation of these regions during a victory outcome. This pattern was then associated with greater alcohol consumption (with regard to lack of mPFC activation) and more addiction symptoms (versus hyperactivation).

### 3.4. E-Chronotype and Suicidality

Some interesting studies have chosen to investigate whether chronotype could be considered among the suicide risk factors. In a sample of 1332 Chinese students aged 12 to 13, an association was found (*p* = 0.005) between E-chronotype and SI/SB (assessed with the Chinese Child Morningness–Eveningness Scale—CMES and with the Child Behavioural Checklist—CBCL, respectively) [73]. In addition, in a population of 89 suicide attempters, E-chronotype, assessed using MEQ, was found to be linked (*p* < 0.05) to impulsiveness, assessed using the Barratt Impulsiveness Scale (BIS), which was also correlated (*p* < 0.05) with more violent suicide attempts (SA) [74]. Despite these findings in both adult and child populations, the univocal and direct association between E-chronotype and suicidality does not seem to be easily demonstrable with enough clarity. However, in line with the above, some possible mediators of the potential correlation could be highlighted. Seasonality and climate changes are well-known characteristics of the epidemiology of suicide [75,76,77,78,79,80,81,82,83], unipolar depressive recurrence [84,85], BD I, BD II, and seasonal affective disorder (SAD) recurrency [86]. As pointed out earlier, mood swings, both in psychiatric and non-psychiatric patients, also occur seasonally and may be further promoted by circadian dysregulation towards eveningness. In addition, late chronotype may facilitate affective destabilizations, though whether this is through biological or psychosocial mechanisms is unclear. In a sample of 120 adult patients affected by MDD, SI scores were higher in patients with seasonality, E-chronotype (*p* < 0.001), and hypomanic personality traits [87], thus illustrating that chronotypes and seasonality [88] may be implicated in the correlation of MDD and SB. A 2018 review taking into account 13 studies with a total sample of 3529 subjects (non-psychiatric, MDD, BPD I-II, schizophrenia, Post-Traumatic Stress Disorder, Borderline Personality Disorder, suicidal) tries to illustrate the complex relationship that seems especially verified in unipolar depression between seasonality, decreased rhythmicity, eveningness, and suicidality [89]. Although chronotype alone did not appear to be significantly associated with suicide (*p* = 0.139), by considering the indirect effect of the depressive symptoms, the correlation became significant (*p* = 0.009) in a sample of 5632 university students in which depressive symptoms, chronotype, and suicidality were systematically assessed [90]. In addition to depression, psychological pain, or psychache, also shows a role as a potential mediator between eveningness and suicidality [91].

Importantly, the mediating element between chronotype and SI/SB may also be represented by insomnia [53,54] by leading to poor daytime functioning and negative affectivity outcomes.

Finally, recalling the previous section, the intensities of mood symptoms could also promote the association between chronotype and suicide, which was not found to be direct.

### 3.5. E-Chronotypes and Suicidality: Immune Dysregulation as a Further Possible Bridging Factor?

Eveningness seems to be associated with greater local and systemic inflammation. In fact, displaying an E-chronotype is linked to a greater risk for Inflammatory Bowel Disease (IBD) [33], asthma [34], cardiovascular disease [35], metabolic disease [36], and other immune-mediated diseases [37], as well as higher blood levels of inflammatory cytokines without any clinical manifestation [92]. At the same time, neuroinflammation and systemic inflammation are associated with SI/SB both through and independent of the presence of a depressive condition [93]. The complex pathophysiology of suicide is the object of a review by Brundin and colleagues [94], which illustrates how diverse inflammatory conditions (traumatic brain injury, vitamin deficiency, autoimmune disorders, and infection) can lead to the hyperactivation of the hypothalamic–pituitary–adrenal axis and to alterations in the monoamine pathways) by dysregulating the kynurenine pathway of tryptophan metabolism. As it is known, monoamine neurotransmission is involved in the development of depressive symptoms that again appear central as a mediating factor in the suicide–inflammation correlation. Thus, the study contributes to providing a well-structured and valid biological model that could possibly explain how an immunological hyper-activation could be involved in the development of mood symptoms and in suicide risk. Serafini et al. [95] conducted another revisional study underlining a possible association between SB and inflammatory cytokine abnormalities. A review by Vasupanrajit et al. [96,97] also associates SI/SB with immune-inflammatory and nitro-oxidative pathway activation, which, by stimulating macrophages to secrete cytokines (IL-1β, IL-6, and TNF-α) and T-cells to secrete INF-γ, increase the level of C-Reactive Protein (CRP). Blood levels of IL-6 and TNF-α (not IL-1β, which does not emerge as significant) also appear to be higher in patients with SB than in healthy controls (*p* = 0.0001 and *p* = 0.01, respectively) in a meta-analysis conducted by Neupane and colleagues [98]. These products exert a neurotoxic effect that could be involved in the manifestation of SI/SB. Additional biological pathways linking suicide to hyperinflammation have recently been proposed by Costanza et al. [99], who investigated the eventual link between the general hyper-inflammatory condition following COVID-19 infection and increased SI/SB in infected patients, as well as the eventual role of anti-inflammatory molecules that, by deregulating dopamine neurotransmission, may represent a further risk factor [100]. Furthermore, as illustrated by Donegan et al. [101], in blood samples of suicidal patients, increased levels of pro-inflammatory cytokines, including, in particular, IL-6 (which also appears to be increased in cerebrospinal fluid), IL-1β, and human C-Reactive Protein (hCRP), are frequently observed. Thus, suicide appears to be correlated with higher inflammation levels both in the CNS (as indicated by the analysis of cerebrospinal fluid) and in the periphery. Regarding peripheral inflammation, Aguglia et al. [102], in study conducted among a sample of 432 hospitalized suicide attempters, found a higher level of CRP within high-lethality cases and also among suicide re-attempters [103]. Interestingly, the aim of this work was to set a cut-off of CRP that should have a significant predictive value for a high-lethality suicide attempt. The result indicated CRP > 4.65 mg/L as corresponding to a sensitivity of 68% and a specificity of 71%.

A further step in exploring an eventual direct association between inflammation and SI (without any mediating role of depressive symptoms) was made in a cross-sectional study by Bergmans et al. [104] involving four indicators of inflammation: circulating levels of CRP, white blood cell (WBC) count, immunoglobulin-E (Ig-E), and dietary inflammatory potential (measured using the Diet Inflammatory Index (DII)). That study, by assessing SI and MDD with the Patient Health Questionnaire-9 (PHQ-9), distributed participants into four groups (SI with MDD, SI without MDD, MDD without SI, and neither SI nor MDD) and did not find any inflammatory index capable of distinguishing SI in MDD. The most interesting result was a direct association between DII and SI in individuals who did not present MDD. A case-control study by Fernendez-Sevillano et al. [105] showed a similar result. The sample of 96 individuals (20 controls and 76 affected by MDD) was divided into subgroups: MDD patients with a recent suicide attempt, MDD patients with a lifetime history of suicide attempts, MDD non-attempters, and controls. Blood levels of IL-2, IL2-R, IL-4, IL-6, and TNF-α were measured in all of the participants. Significantly, IL-6 concentrations were higher in both recent (*p* = 0.04) and distant (*p* = 0.015) attempters in comparison to MDD non-attempters. Thus, the plasma level of this cytokine appears to be correlated to SB without any association to MDD.

An additional factor that we have to underline is the possible involvement of chronotype in the determination of inflammatory states, as illustrated in a cross-sectional study conducted by Orsolini and colleagues [106]. Among 133 retrospectively recruited patients with moderate–severe depression (both unipolar and bipolar), higher levels of baseline high-sensitivity CRP were associated with SA (*p* = 0.05), death (*p* = 0.018), and self-harm/self-injury thoughts (*p* = 0.011). In addition, depressive temperament emerging from the TEMPS scale and lower MEQ scores (i.e., eveningness) statistically significantly predicted higher hsCPR blood levels [106]. Thus, inflammation appears to be positively correlated to SI/SB, depressive temperament, and E-chronotype. As underlined by discussing other studies, several biologic theoretical models try to explain the link between inflammation and SI/SB. By considering E-chronotype as a risk factor for numerous inflammatory conditions, we may find one of the possible bridging points between chronotype and suicide.

## 4. Discussion

Taking into account the most recent and relevant literature, the present review did not find any direct or significant correlation between E-chronotype and SI/SB. Nevertheless, eveningness, which potentially leads to SJ and circadian disruption [7,39,40], may represent a bio-personological, relatively stable trait involved in the development of a higher degree of social anxiety [49], mood destabilization [56,57,61], emotional dysregulation [49], impulsivity [47,49,52], aggression [48,49], alterations in the reward functions [3], and insomnia [50,51,52,53,54]. All of these psychopathological dimensions may play a central role in promoting the further crystallization of more defined psychiatric conditions, such as mood, personality, and substance use disorders, and, finally, the expression of self-harm and SI/SB. In addition, the clinical course of fully developed affective disorders that worsen in terms of recurrence, complexity of episodes, and more significant levels of emotional dysregulation may exert a mediating effect between E-chronotype and suicide [84,85,86,90], as might the presence of additional comorbidities, such as substance/alcohol use disorders [72] and ADHD [46].

E-chronotype also seems to be associated with hyperinflammatory conditions, both local and systemic [93], and the possibly related neuroinflammation may represent a further bridging point between chronotype and suicide [94,101]. As we tried to illustrate in the dedicated paragraph, the overexpression of some inflammatory pathways could be linked to alterations in neurotransmission, especially by affecting monoamine metabolism, which plays a leading role in the pathophysiology of mood disorders [94]. Thus, we may consider E-chronotype, perhaps through the additional mediation of systemic/local inflammation, as a risk factor for MDD, BSDs, and SAD, which are all notoriously associated with SI/SB. Of course, pointing out distinct molecules that could serve as markers for assessing suicide risk is not so immediate. In fact, findings of immune dysregulation in suicidal patients are not uniform, but, in the literature that we analyzed, IL-6 and CRP were the most significantly altered inflammatory parameters [101,102,105,106]. Moreover, studies that show the possibility of a direct correlation between inflammation and suicide (without the mediating effect of depressive symptomatology) are maybe the most interesting from our point of view [104,105]. In fact, they may establish a new linking perspective on the correlation between E-chronotype and suicide, which already appears to be mediated by depressive symptoms but could perhaps also be bridged by neuroinflammation alone, thus providing another possible path to follow while we try to unravel the complex and intricate role of chronotype in human pathophysiology.

The aim of this study was to narratively summarize the evidence from the recent literature that may set chronotype evaluation as an important tool for clinicians dealing with suicide prevention. In fact, an accurate assessment may guarantee, e.g., the possibility of a focused psychoeducational intervention constituting a precious element for primary and secondary prevention. In addition, the possible correlation between inflammation, chronotype, and SI/SB may also be considered, on one hand, as a new research topic to be explored, and, on the other hand, as a possible instrument for early risk assessment carried out through the use of increasingly sophisticated biomarkers.

The limitations of this study are several. First, the selection method, which was non-systematic, exposes our conclusions to a possible selection bias, which is typically associated with narrative reviews. Next, few studies were selected, illustrated, and discussed, thus presenting just a narrow view of what the literature may contain about this specific topic. Moreover, data are presented in a preliminary sequence summary of results. We must underline again that no clear evidence was found regarding a direct association between chronotype and suicide. Thus, the mediating factors alone may justify the result without specifically involving eveningness. In addition, the selected studies present important elements of heterogeneity, including the adopted methodologies, chronotype assessments, evaluations of psychopathological elements, numerosity, and characteristics of the selected samples (age, gender, geographical provenience, social conditions, health conditions, psychological and psychopathological conditions, etc.). Finally, the quality of the selected studies was not quantified, as recommended for more systematic reviews. The merit of this review may lie in the originality of the overall thesis, which is particularly articulated and requires the construction of a complex chain of causal hypothetical elements.

## 5. Conclusions

The assessment of sleep–wake cycles, which can be conducted with different instruments (e.g., MEQ, CSM, and MCTQ) [7,47,57] and through the evaluation of several biological parameters [42], may be progressively considered as a central point during psychopathological investigation. E-chronotype progressively appears to be a transdiagnostic risk factor for a pool of psychiatric conditions, among which suicidal behaviors stand out. As we tried to underline, later chronotypes could increase the risk of suicide not only by fostering autoinflammatory conditions and patients’ disease burden, but perhaps also by promoting a more fundamental condition of immune dysregulation both systemically and at the CNS level [93,94,101]. Thus, a laboratory quantification of inflammatory markers (RCP, cytokines) may help clinicians to further stratify suicidal risk and the complexity of different clinical presentations. Further systematic studies are needed to investigate the value and the stability of chronotypes.

## Figures and Tables

**Table 1 brainsci-14-00030-t001:** Biological, social, and psychological factors possibly influencing the chronotype expression of individual chronotype.

Possible Influencing Factor	Main Findings
Genetics [12,13,14]	Three different genome-wide association studies (GWAS) found that the genes PER2, RGS16, FBXL13, and AK5 are related to the phenotypical expression of chronotype. The gene hPer2 also seems to be involved.
Age [15,16,17,18]	Eveningness is most expressed during adolescence and young adulthood. There is substantial flattening of differences between males and females during the post-menopausal period, suggesting the importance of endocrinological factors.
Light Exposure [19,20,21,22,23]	Light is the strongest *zeitgeber* for the circadian system.
Others [22]	Level of education, religiosity, and cohabiting with small children (possible mediating effect exerted by light exposure).

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
