# Peer review of "Evening Chronotype and Suicide: Exploring Neuroinflammation and Psychopathological Dimensions as Possible Bridging Factors—A Narrative Review"

_brainsci, 2023, doi:10.3390/brainsci14010030_

Round 1
Reviewer 1 Report (Previous Reviewer 3)
Comments and Suggestions for Authors
Overall, I agree with the authors' claim in the end of their reply that "additional changes, the manuscript will become more fluid for the reader
and our hypothesis will emerge clearly"
Author Response
We thank the reviewer for the comments
Reviewer 2 Report (Previous Reviewer 2)
Comments and Suggestions for Authors
The authors have addressed my queries adequately and necessary changes have been made in the manuscript.
Comments on the Quality of English LanguageNo issues detected.
Author Response
We thank the reviewer for the comments
Reviewer 3 Report (Previous Reviewer 1)
Comments and Suggestions for Authors
Most comments have been addressed in this version of the manuscript.
Still, the abstract should follow the style of structured abstracts, but without headings, so please delete Background, Objective, and other subheadings from your abstract.
Comments on the Quality of English LanguageThere are some typos and grammar issues in the text, and it needs proofreading (e.g. last sentence of the abstract).
Author Response
We thank the reviewer for the comments. We will provide a new revised version of the manuscript without headings in the abstract section
This manuscript is a resubmission of an earlier submission. The following is a list of the peer review reports and author responses from that submission.
Round 1
Reviewer 1 Report
Comments and Suggestions for Authors
Formatting
- The abstract should follow the style of structured abstracts, but without headings, so please delete (1) Background and other subheadings from your abstract.
Methods:
Inclusion criteria are not specified. How did you select the articles? Were there any restrictions regarding methodology, age of participants or sample sizes, for example? This is somewhat discussed in the last paragraph of the article, but the reader expects this information in the methodology section.
To better illustrate the scope of work, I can also recommend providing general information on how many articles were screened and how many were included in the final review. Also, how many papers were original articles and how many reviews?
Results:
3.1 The E-chronotype and social jetlag section describes only the studies on young people and children. Is there any relevant research on adults?
Sections 3.2 and 3.3 could be restructured for clarity according to disorders/states (depression, BD, reward function). As it is now, both sections discuss the links and likelihood of connections between chronotypes and psychiatric conditions, so it is hard to compare studies referring to the same condition. It would be beneficial to highlight the relationship between depression (as the leading cause of suicide) and chronotypes/inflammation.
3.4., as in 3.1, does not separate studies in children and adults, although the two age groups have distinctive sleep-wake cycles.
Are studies 74 and 75 the only ones that directly assessed the connection between suicide and chronotypes? If so, it would be beneficial to highlight this research gap. Otherwise, some more generalized data is essential.
Line 230: Unclear sentence: “In addition, the late chronotype may facilitate affective destabilizations either through the same sea sound or through different mechanisms”.
Section 3.5 provides substantial evidence linking SB to neuroinflammation and systemic inflammation. However, the studies on the association of E-chronotype with inflammation are scarce. The study by Orsolini has depression as a strong mediating factor, so it shows the involvement of chronotypes only indirectly. It is understandable that a scoping review cannot cover all available literature, but this specific seems central to this article and requires special attention.
The authors do conclude that ‘the present review did not find any direct and significant correlation between E-chronotype and SI/SB’, which aligns with the data presented. However, they insist on seeing chronotypes as a biological trait underlying the development of a range of psychophysiological disorders, potentially leading, in their term, to SB. Therefore, it would be beneficial for the review to provide short summaries (1-2 sentences) regarding each major disorder and its association with the E-chronotype. This would help create a more coherent story, linking chronotypes, inflammation and suicide.
Comments on the Quality of English LanguageRegarding the quality of English, some proofreading is required. The authors could also improve style by chunking long sentences that are hard to follow (e.g., lines 258-264). Overall, the text has no significant language-related issues.
Reviewer 2 Report
Comments and Suggestions for Authors
Comments:
This article is a scoping review about Evening -chronotype and suicide. The research question is clinically important. However, there are some concerns about the methodology. I doubt, if it is really a “Scoping review”. Many parts of the review do not adhere to the guidelines for scoping review. I have enumerated them below which the authors need to address:
1- Was the protocol for this scoping review registered at any database? If yes, the registration number should be shared.
2- What was the inclusion criteria adopted for this scoping review? What were the search strategies adopted? Only mentioning use of ‘Indexed terms’, free text terms’ is not adequate.
3- Search strategies of at least one database may be provided as supplementary material.
4- How the data was screened? Which data were extracted? How they were extracted? How the disagreements were resolved? These points should be mentioned clearly in the methods section.
5- No assessment of quality for the included study has been done. Mentioning the lack of it as ‘limitation” is not sufficient. There are no valid reasons for not conducting the quality assessment.
6- There is no information about the number of articles searched and final number of articles included for this scoping review. No PRISMA diagram has been provided.
7- The authors have described different aspects of E-chronotype and suicide. Relevant articles on the subject have been referred and interpretations are adequate.
8- In my opinion, the article does not adhere to the guidelines for scoping review. It may qualify as a narrative/ literature review.
Comments on the Quality of English Language
Minor editing of the English language required.
Reviewer 3 Report
Comments and Suggestions for Authors
The authors submitted a scoping review of relation between evening chronotype and suicide and proposed a hypothetical mediating role of neuroinflammation.
Major concern.
The information provided in this submission on the reviewing process is insufficient. To remind, scoping reviews define eligibility criteria, search the literature, screen the results and select evidence for inclusion. It is necessary to follow the guidelines for scoping review, e.g.:
https://www.ncbi.nlm.nih.gov/pmc/articles/PMC9580325/#:~:text=A%20scoping%20review%20is%20a,literature%20on%20a%20given%20topic.
When following the guidelines for scoping review, it is necessary to present the extended PRISMA list. The PRISMA extension for scoping reviews is a list of 20 essential reporting items for review teams when completing a scoping review:
http://prisma-statement.org/Extensions/ScopingReviews
Other concerns.
1. Although, in Abstract, “(2) Objective: This scoping review aims to summarize the evidence supporting the possible association between chronotypes and suicide…” this Abstract is lacking Conclusion, and even the results were not reported (it is only written: “(4) Results: Despite the apparent absence of a direct correlation between E-chronotype and suicidality”…
2. Abstract starts with the statement that “chronotype is defined as the variability of the phase angle of entrainment”. This definition is totally wrong, because chronotype is measured by the author of such a definition with a questionnaire, without conducting any measurements of the phase angle between the environmental cycle and a marker of phase of the circadian clocks, e.g., melatonin onset or core body temperature minimum, and the phase angle between this marker and the sleep-wake cycles also remains unknown.
3. The title is misleading because the fist sentence of Discussion says “Taking into analysis most recent and relevant literature, the present review did not find any direct and significant correlation between E-chronotype and SI/SB”(“ suicidal ideation (SI) and behavioral (SB)”).